# INVERSE-AND-EDIT: SIMPLE AND EFFECTIVE FRAMEWORK FOR FAST IMAGE EDITING

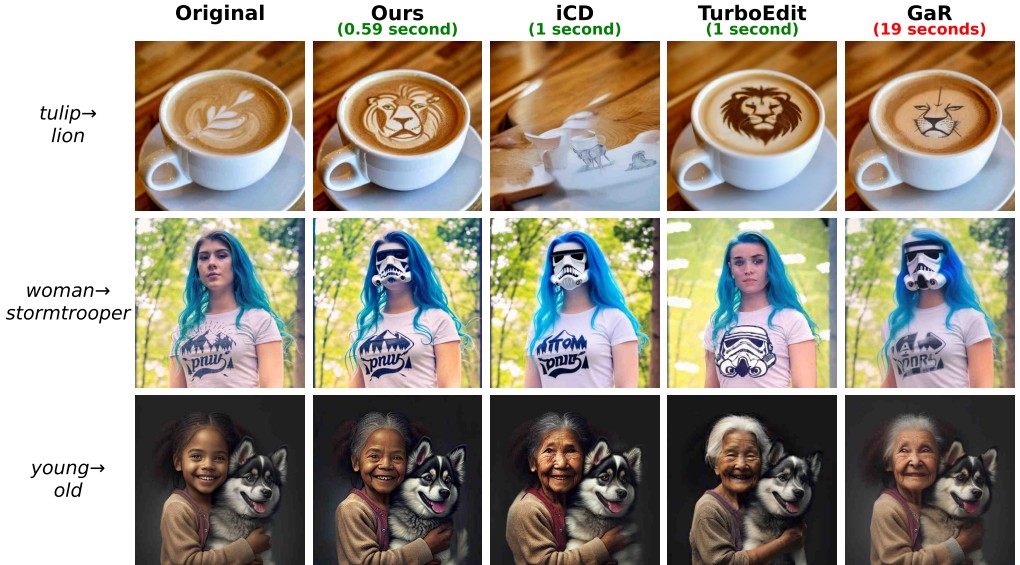

Figure 1: Examples of image editing results obtained using various pipelines, including our method. Our approach enables accurate editing with precise detail preservation and sub-second processing time.

## ABSTRACT

Recent advances in diffusion-based image editing have achieved impressive results, offering fine-grained control over the generation process. However, these methods are computationally expensive due to their iterative nature. While distilled diffusion models enable faster inference, their editing capabilities remain limited - primarily because of poor inversion quality. High-fidelity inversion and reconstruction are essential for precise image editing, as they preserve the structural and semantic integrity of the source image. In this work, we propose a simple, general framework that optimizes the diffusion model over the entire inversion and generation trajectory and is compatible with arbitrary accelerated diffusion backbones, enabling high-quality editing in under one second. We achieve state-of-the-art performance across various image editing tasks, accelerated diffusion models, and datasets, demonstrating that our method matches or surpasses full-step diffusion models while being substantially more efficient.

## 1 INTRODUCTION

Diffusion-based generative models (Ho et al., 2020; Song et al., 2021) are widely adopted for text-to-image generation, owing to their stable training dynamics, strong mode coverage, and high-fidelity, diverse samples. Beyond their use in synthesis, diffusion models have been adapted to practical tasks that span interpretability and controllability, personalized generation, and discriminative analysis using diffusion-derived features. A key downstream task is text-guided image editing: given a source

image and a textual description of the desired modification, the model generates an updated image by steering the iterative sampling trajectory to apply fine-grained, prompt-conditioned changes while preserving source content.

Full-step diffusion pipelines achieve strong image editing quality but require substantial inference time. To address this latency, recent work explores distilled diffusion models. The backbones of such methods can be broadly categorized into two groups: consistency-based (Song et al., 2023; Ren et al., 2024) and fast generator-based approaches (Sauer et al., 2024; Yin et al., 2024). Consistency-based methods preserve a solver-based view of diffusion dynamics by optimizing few-step solvers for the underlying differential equation (e.g., Latent Consistency Models (Luo et al., 2023), Hyper-SD (Ren et al., 2024), Invertible Consistency Distillation (Starodubcev et al., 2024)), whereas generator-based models (e.g., SDXL-Turbo (Sauer et al., 2023)) train a neural network $G_\theta$ to map standard Gaussian noise $z$ directly to high-quality images in a few steps. Focusing on fast editing, we observe that editing algorithms vary widely and exhibit strong backbone dependence. For example, InfEdit (Xu et al., 2023b) demonstrates impressive results within consistency models but is effectively restricted to LCM–DreamShaper (Luo et al., 2023) due to its reliance on a stochastic consistency sampler. Meanwhile, an increasing number of consistency backbones adopt deterministic, trajectory-segmented sampling (e.g., Hyper-SD (Ren et al., 2024)), which makes direct reuse of the InfEdit design non-trivial. Consequently, existing fast editing methods generalize poorly across accelerated backbones and often require additional adaptation.

Unlike full-step diffusion methods – where direct backpropagation through the entire inversion and generation pipeline is computationally infeasible – our approach is expressly designed for few-step models, and we demonstrate its effectiveness. In this work, we propose a simple and generalizable framework for accelerated image editing, compatible with a broad range of distilled backbones, without relying on backbone-specific internals or solver assumptions. Our approach builds on external forward consistency model training that approximates the noising dynamics of an accelerated backbone. We further introduce a cycle-consistency loss to improve inversion quality and preserve semantic details. We apply the framework to a broad set of different accelerated methods and evaluate its effectiveness on PIE-Bench. Our main contributions are as follows:

- We introduce a lightweight framework for accelerated image editing based on forward Consistency Model training, easily adaptable to different distilled backbones.

- Our method enables robust and transferable editing across a wide spectrum of accelerated diffusion models: DreamShaper-LCM (Luo et al., 2023), SDXL-Turbo (Sauer et al., 2023), Hyper-SD (Ren et al., 2024), iCD (Starodubcev et al., 2024).

- We achieve state-of-the-art accelerated editing results on PIE-Bench, outperforming backbone-specific approaches while being significantly faster and requiring no extra inference as in Prompt-to-Prompt or MasaCTRL.

## 2 RELATED WORK

Diffusion-based approaches are widely used for image generation due to strong pretrained priors that support diverse, high-quality semantics, which also makes them effective for text-guided image editing. Editing methods balance incorporating information from a target prompt with preserving alignment to the source content. They are often grouped into three (partly overlapping) categories: optimization-based, attention manipulation, and guidance-driven methods.

**Editing with full-step diffusion models** Most full-step methods begin with inversion, i.e., approximating a latent state of the source image at a chosen timestep using a pretrained model. This state then initializes a new, target prompt-conditioned sampling trajectory. Optimization-based approaches (Miyake et al., 2024; Mokady et al., 2022) perform per-sample inversion optimization, which involves additional, computationally expensive iterations during the editing process. These methods improve inversion quality by optimizing prompt embeddings. Attention-based approaches (Cao et al., 2023; Hertz et al., 2022) demonstrate strong performance but may lack fine-grained controllability. Prompt-to-Prompt operates on cross-attention by preserving maps for tokens shared between source and target prompts and adjusting attention maps accordingly. For shared tokens, the original maps from the source inference are retained, for new tokens, the maps are updated

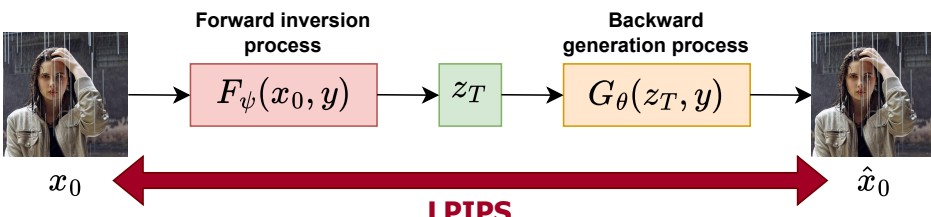

Figure 2: Diagram of our cycle-consistency loss. We optimize the forward consistency model by backpropagating through the image reconstruction process. A patch-wise LPIPS loss is used to enforce perceptual similarity between the original and reconstructed images

to reflect the target prompt. This method often requires carefully selected blend words to enable effective editing.

**Editing with accelerated diffusion models** Distilled backbones reduce sampling cost but may degrade trajectory fidelity, making standard inversion less reliable. In some cases, deterministic inversion is not guaranteed under stochastic sampling or non-invertible updates. For instance, InfEdit (Xu et al., 2023b), built on a DreamShaper-LCM backbone, adopts a virtual inversion scheme that combines MasaCTRL (Cao et al., 2023) with Prompt-to-Prompt (Hertz et al., 2022). GNRi (Samuel et al., 2025) and PostEdit (Tian et al., 2025) employ energy-based guidance during editing, which introduces additional iterations and can erode the speedups. Invertible Consistency Distillation (Starodubcev et al., 2024) trains separate forward and backward models for inversion and generation, which are then combined with Prompt-to-Prompt to improve content preservation. The method relies on joint distillation of two models, constraining training to a simple, fixed procedure and thereby may limit compatibility of richer modern distillation methods such as Hyper-SD. Finally, the diversity of distilled backbones has yielded a landscape of editing methods that are often not directly compatible across backbones.

## 3 PRELIMINARIES

**Diffusion model** Our method builds on a latent diffusion text-to-image model (LDM) that encodes images into a low-dimensional space with a variational autoencoder (VAE). Let $z_0$ be the VAE latent of an image, and let $\{\alpha_t\}_{t=1}^T$ be a variance schedule with cumulative product $\bar{\alpha}_t = \prod_{s=1}^t \alpha_s$. The forward noising process is

$$z_t = \sqrt{\bar{\alpha}_t}\, z_0 + \sqrt{1 - \bar{\alpha}_t}\, \varepsilon, \qquad \varepsilon \sim \mathcal{N}(0, I).$$

The denoiser $\varepsilon_\theta$ is trained to predict the noise by minimizing: $L(\theta) = \mathbb{E}_{t, z_0, \varepsilon}[\|\varepsilon_\theta(z_t, t, y) - \varepsilon\|_2^2]$, where the text condition $y$ is injected via cross-attention. After training, samples are generated by iteratively denoising from $z_T \sim \mathcal{N}(0, I)$ with the update

$$z_{t-1} = \sqrt{\tfrac{\bar{\alpha}_{t-1}}{\bar{\alpha}_t}}\, z_t + \varepsilon_\theta(z_t, t, y)\left(\sqrt{1 - \bar{\alpha}_{t-1}} - \sqrt{\tfrac{\bar{\alpha}_{t-1}}{\bar{\alpha}_t}}\sqrt{1 - \bar{\alpha}_t}\right) \tag{1}$$

**Distilled diffusion models** Since inference time remains the primary limitation of diffusion models, a wide variety of distillation techniques have been proposed. We group them into two categories: generator-based methods (e.g., SDXL-Turbo (Sauer et al., 2023)) and consistency-based methods (e.g., LCM (Luo et al., 2023), Hyper-SD (Ren et al., 2024)). Generator-based methods train a student that samples in a few steps to imitate a teacher that uses many steps via noise or $x_0$ regression, sometimes with perceptual or adversarial components. Consistency-based methods view sampling through the probability flow ODE and enforce self consistency across timesteps, so that valid samples arise in $4 - 8$ steps.

**Diffusion Editing** Diffusion editing comprises two stages: inversion and generation. Inversion estimates a latent $z_T$ for the source image in the latent space of a diffusion model; generation then applies the target prompt while using $z_T$ as the initial point for denoising. In practice, we perform

inversion using the deterministic reverse DDIM (Song et al., 2022) update obtained by rearranging Eq. 1 and resolving the implicit dependence via the standard adjacent-step approximation yields the practical rule in Eq. 2.

$$z_t \approx \sqrt{\frac{\bar{\alpha}_t}{\bar{\alpha}_{t-1}}} z_{t-1} - \left( \sqrt{\frac{\bar{\alpha}_t(1-\bar{\alpha}_{t-1})}{\bar{\alpha}_{t-1}}} - \sqrt{1-\bar{\alpha}_t} \right) \varepsilon_\theta(z_{t-1}, t, y) \tag{2}$$

The resulting representation $z_T$ then serves as the starting point for generation, conditioned on a target prompt that specifies the desired edits. Consequently, many full-step approaches regard inversion as sufficiently accurate and optimize primarily the generation process. To improve editing quality, various approaches have been proposed, including optimization-based techniques (Mokady et al., 2022), attention manipulation methods (Cao et al., 2023; Hertz et al., 2022), and guidance-driven strategies (Bansal et al., 2023; Titov et al., 2024).

**Accelerated diffusion editing approaches** Compared to full-step diffusion, few-step samplers take large jumps along the teacher trajectory, so the adjacent-step assumption used by DDIM-style inversion ($z_t \approx z_{t-1}$) often breaks, turning inversion into a key bottleneck for high-quality editing. To adapt techniques originally developed for full-step pipelines, accelerated methods refine the inversion stage in different ways. Existing fast editing approaches are highly diverse and pursue quality through different mechanisms, which often makes them sensitive to the chosen backbone and limits scalability. We categorize editing methods by the base model they are built on: On generator-based backbones, GNRi (Samuel et al., 2025) and ReNoise (Garibi et al., 2024) refine the process of inversion by adding per-sample optimization iterations, whereas TurboEdit alters the sampling via time-shifting and a Delta Denoising score (Hertz et al., 2023) update, reducing reliance on explicit inversion. As for consistency-based approaches, InfEdit adopts a Latent Consistency Model distilled from DreamShaper (Luo et al., 2023) and generates images using a stochastic consistency sampler. It also employs a virtual inversion framework, to enable editing without true inversion. The only exception is Invertible Consistency Distillation(iCD) (Starodubcev et al., 2024), which distills from a teacher that uses the full sampling schedule. It shares the same consistency-model formulation but jointly trains a forward consistency model for inversion and a backward consistency model for generation under a multi-step consistency objective. Since inversion is then provided by the forward consistency model, the need for virtual inversion disappears. Consistency models approximate a function $f_\theta$ that maps a noisy point $z_t$ at any timestep $t$ of the diffusion ODE trajectory to its origin $z_0$. Given a pretrained teacher diffusion model $\epsilon_\eta$, this is achieved through the consistency distillation objective:

$$\mathcal{L}_{CD}(\theta) = \mathbb{E}[d(f_\theta(z_{t_{n-1}}, t_{n-1}), f_\theta(z_{t_n}, t_n))] \to \min_\theta, \tag{3}$$

where $z_{t_{n-1}}$ is obtained by applying one solver step of the teacher model. This loss enforces the self-consistency property: $f_\theta(z_{t'}, t') = f_\theta(z_t, t); \quad \forall t \in [t_0, t_N]$. Invertible Consistency distillation (Starodubcev et al., 2024) demonstrates that image inversion can be achieved using a forward consistency model, which is trained jointly with a backward consistency model. The ODE trajectory is divided into multiple segments: the forward model is trained to map any point within a segment to its final boundary, while the CM maps it to the starting boundary. The consistency distillation loss from Equation 3 is adapted for both the forward and backward consistency models training objectives and is combined with additional symmetrical preservation losses for forward ($\mathcal{L}_f$) and backward ($\mathcal{L}_r$) models, defined as follows:

$$\mathcal{L}_f(\theta, \psi) = \mathbb{E}\left[ d\left( f_\theta(g_\psi(z_{s_t})), \ z_{s_t} \right) \right] \to \min_\theta \tag{4}$$

Recent consistency-distilled backbones with deterministic, trajectory-segmented sampling (e.g., Hyper-SD (Ren et al., 2024)) are not plug-and-play for either InfEdit (Xu et al., 2023b) or iCD (Starodubcev et al., 2024): the first one relies on stochastic consistency sampling and virtual inversion, whereas the second one presupposes joint training of forward and backward consistency models with access to teacher. Consequently, porting these pipelines to a pretrained Hyper-SD checkpoint is feasible in principle but operationally nontrivial. In practice, this typically requires sampler changes (for InfEdit) or retraining bidirectional models (for iCD), which motivates a backbone-agnostic editing framework.

## 4 METHOD

Our task is developing a universal training procedure for an image inversion model. In contrast to iCD (Starodubcev et al., 2024), we train the noising (forward) model for any accelerated pretrained backbone. Taking advantage of the fact that distilled models require far fewer model evaluations than full-step teachers, we train efficiently through the entire inversion and generation loop with an accelerated diffusion model, demonstrating accurate reconstructions and compatibility across backbones and schedules. Unlike prior approaches such as InfEdit or iCD, our framework does not require stochastic samplers, internal solver access, or joint distillation of both forward and backward modules. It supports both plug-in adapters (by training only forward consistency model) and joint fine-tuning (by optimizing both forward and backward consistency models), depending on the available backbone constraints.

We build our approach on the theoretical basis of consistency models. Since it preserves the ODE-trajectory, that is important for inversion-nature of our task and can be easily adapted to any type of backbones. We parametrize the forward $f_\psi$ and backward $g_\theta$ consistency models with LoRA adapters, denoted $\psi$ and $\theta$, respectively. For the backward adapter $\theta$, we use a lower rank and fewer target modules, than for the forward adapter, resulting in a significantly smaller adapter.

### 4.1 GENERAL FRAMEWORK FOR IMAGE INVERSION

We discretize the trajectory into $M$ noise levels $t_0 < t_1 < \cdots < t_M$, defining segments $s_m = [t_{m-1}, t_m]$. For each segment, we train two models:

- **Forward consistency model** approximates a function $f_\psi(z_{t_{m-1}}, t_{m-1}, y)$ maps a latent at timestep $t_{m-1}$ to $t_m$ (noising).
- **Backward consistency model** approximates a function $g_\theta(z_{t_m}, t_m, y)$ maps a latent at timestep $t_m$ to $t_{m-1}$ (denoising).

Given an image $x_0$, we obtain its latent $z_0 = \mathrm{enc}(x_0)$ via a pretrained VAE encoder. The forward function $F_\psi$ applies the forward model $M$ times to produce a final noised latent $z_M$, while the backward function $G_\theta$ denoises $z_M$ back to $\hat{z}_0$ and reconstructs $\hat{x}_0$ using the VAE decoder. So, the full inversion process is the following:

$$z_{t_0} = \mathrm{enc}(x_0), \qquad z_{t_m} = f_\psi(z_{t_{m-1}}, t_{m-1}, y), \quad m = 1, \ldots, M,$$

$$z_{t_M} = F_\psi(x_0, y) \equiv f_\psi\Big(f_\psi\big(\cdots f_\psi(\mathrm{enc}(x_0), t_0, y), t_1, y\big) \cdots, t_{M-1}, y\Big). \tag{5}$$

And the symmetric — generation process defined as follows:

$$\hat{z}_{t_M} = z_{t_M}, \qquad \hat{z}_{t_{m-1}} = g_\theta(\hat{z}_{t_m}, t_m, y), \quad m = M, \ldots, 1,$$

$$\hat{x}_0 = G_\theta(z_{t_M}, y) \equiv g_\theta\Big(g_\theta\big(\cdots g_\theta(z_{t_M}, t_M, y), t_{M-1}, y\big) \cdots, t_1, y\Big). \tag{6}$$

To ensure accurate inversion, we optimize $f_\psi$ and $g_\theta$ through the entire process of image reconstruction since the number of steps $M$ is sufficiently small, defining following **cycle-consistency loss**:

$$\mathcal{L}_{\mathrm{rec}}(x_0) = \mathrm{LPIPS}(G_\theta(F_\psi(x_0, y_{\mathrm{src}}), y_{\mathrm{src}}), x_0) \tag{7}$$

We also present the diagram our cycle-consistency loss in Figure 2. In addition, we incorporate standard consistency distillation losses (Song et al., 2023) and forward preservity loss (Starodubcev et al., 2024).

- $\mathcal{L}_{\mathrm{CD}}(\theta)$ and $\mathcal{L}_{\mathrm{CD}}(\psi)$ (see Eq. 3) ensure each model approximates forward and backward teacher ODE trajectories.
- A forward preservation loss $\mathcal{L}_{\mathrm{f}}(\theta, \psi)$ (see Eq. 4) for $f_\psi$ encourages local compatibility between forward and backward consistency models at each segment.

The final training objective becomes:

$$\mathcal{L} = \mathcal{L}_{\mathrm{rec}} + \mathcal{L}_{\mathrm{CD}}(\psi) + \mathcal{L}_{\mathrm{CD}}(\theta) + \mathcal{L}_{\mathrm{f}}(\theta, \psi) \tag{8}$$

## 4.2 Training Regimes Across Backbones

Depending on the backbone, we either fix $\psi$ (e.g., for pretrained iCD) or jointly optimize $\theta$ and $\psi$ (e.g., for SDXL-Turbo and Hyper-SD). This flexibility enables our method to scale across pretrained diffusion models without requiring architectural changes or retraining from scratch. We also define a pretrained teacher model $\varepsilon_\eta$, which we use in the consistency distillation losses $\mathcal{L}_{CD}$.

**DreamShaper-LCM.**

*Initialization:* $f_\psi$ = DreamShaper-LCM; $g_\theta$ = DreamShaper-LCM; $\varepsilon_\eta$ = DreamShaper.

Both the forward and backward models are initialized from a pretrained DreamShaper-LCM checkpoint and fine-tuned jointly under our objective 8. The original DreamShaper model serves as the teacher. While DreamShaper-LCM was originally designed for stochastic sampling, we adapt it to deterministic, trajectory-segmented inference, enabling compatibility with our framework.

**SDXL-Turbo.**

*Initialization:* $f_\psi$ = SDXL-Turbo; $g_\theta$ = SDXL-Turbo; $\varepsilon_\eta$ = SDXL Base.

Although SDXL-Turbo lacks a formal ODE trajectory, we find that its generative distribution is sufficiently structured to allow training of an forward consistency model. We fine-tune both $f_\psi$ and $g_\theta$ using SDXL Base as the teacher, demonstrating that our method is applicable even to adversarially-distilled models.

**Hyper-SD.**

*Initialization:* $f_\psi$ = Hyper-SD; $g_\theta$ = Hyper-SD; $\varepsilon_\eta$ = Hyper-SD.

As no inversion-compatible methods exist for Hyper-SD, we apply our full joint training pipeline. All models ($f_\psi$, $g_\theta$ and teacher $\epsilon_\eta$) are initialized from the same pretrained Hyper-SD checkpoint. The fine-tuned $f_\psi$ enables faithful inversion and editability without sacrificing generation quality.

**SD v1.5 (iCD).**

*Initialization:* $f_\psi$ = iCD-forward; $g_\theta$ = iCD-backward; $\varepsilon_\eta$ = SD v1.5.

We initialize both $f_\psi$ and $g_\theta$ from public iCD checkpoints. Because $f_\psi$ and $g_\theta$ are already an accelerated, pretrained consistency-model pair, it suffices to fine-tune only $f_\psi$ to achieve high-fidelity inversion. At the same time, to preserve generation quality, the backward model is frozen.

## 4.3 Editing Procedure

Once trained, our framework supports efficient editing without additional components (e.g., Prompt-to-Prompt (Hertz et al., 2022), MasaCTRL (Cao et al., 2023), or guidance-based optimization (Titov et al., 2024)). Given an image $x_0$, source prompt $y_{src}$ and target prompt $y_{trg}$:

$$z_{t_M} = F_\psi(x_0, y_{src}); \qquad x_{edited} = G_\theta(z_{t_M}, y_{trg})$$

This simple noising-denoising pipeline achieves high-quality, semantically consistent edits while operating in a few steps. In our experiments, it consistently surpasses competing methods built on the same backbone, while being significantly faster, as it avoids additional optimization inferences and uses only a few model calls per edit. Despite the absence of Prompt-to-Prompt blending, our approach improves inversion fidelity and enables high-quality editing. In comparison with baseline iCD method, our approach eliminates the need of additional P2P (see Figure 11).

## 5 Experiments

**Fine-tune setup** Cycle-consistency loss (see Eq. 7) is performed by optimizing the LPIPS objective with a VGG-16 backbone (Simonyan & Zisserman, 2015; Zhang et al., 2018), as it captures

| Original | iCD | Ours | Original | iCD | Ours |
| --- | --- | --- | --- | --- | --- |

Figure 3: Visual comparison of results produced by our fine-tuned model and the baseline. Our method significantly improves image reconstruction, providing samples that are consistent with the original image.

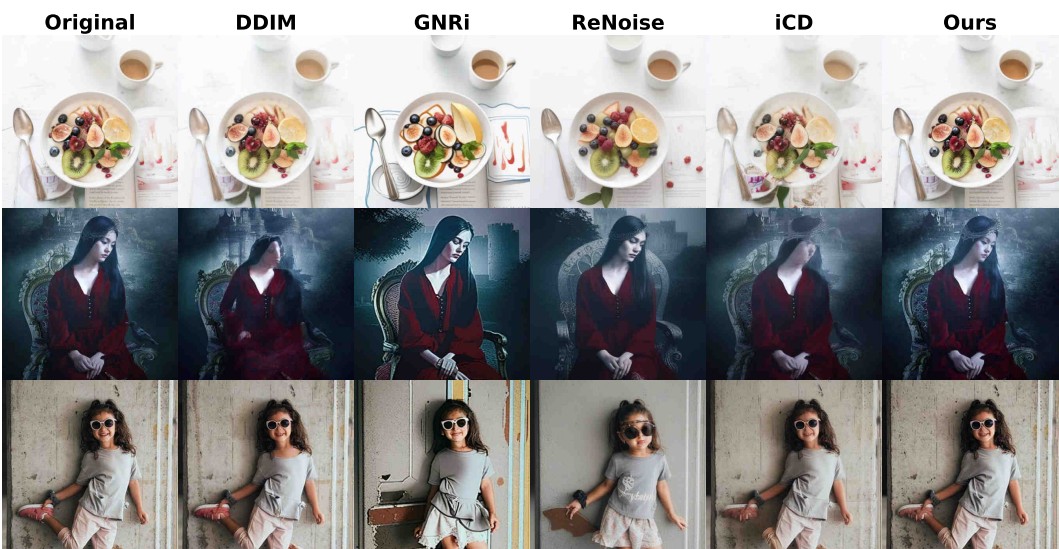

| Original | DDIM | GNRi | ReNoise | iCD | Ours |
| --- | --- | --- | --- | --- | --- |

Figure 4: Examples of image reconstruction obtained from our method and from other approaches. Cycle-consistency loss facilitates accurate image reconstruction, maintaining both global facial structure and local detail.

structural and perceptual differences relevant to image reconstruction. Images are divided into nine 224x224 patches to match the VGG-16 training setup. We optimize $\psi$ and $\theta$ (if it is required by backbone) parameters. To keep local consistency properties within each segment, we also retain the forward preservation loss $\mathcal{L}_f$, along with the consistency distillation loss, to enforce the consistency properties of a forward model. In setups where the backward model is also fine-tuned, we additionally include a reverse-consistency loss $\mathcal{L}_{CD}(\psi)$ and update $\psi$ jointly. Fine-tuning is conducted on the training split of MS-COCO (Lin et al., 2015) and evaluated on the validation split.

**Inversion and editing setup** We evaluate inversion and editing performance on multiple datasets. For inversion experiments, we use Pie-Bench (Ju et al., 2023) for qualitative evaluation, and more than 2700 high-resolution images from the MS-COCO (Lin et al., 2015) for quantitative evaluation. For image reconstruction, we use classifier-free guidance equal to zero for all methods (see Appendix A.2). We use samples from MS-COCO for quantitative evaluation, as it contains a large amount of labeled data at various scales, enabling the detection of significant differences between methods. For editing experiments, we use 420 images from publicly available Pie-Bench, following Starodubcev et al. (2024), which includes a broad range of edit types for qualitative and quantitative evaluation. Like iCD (Starodubcev et al., 2024), we adopt a dynamic classifier-free guidance (CFG), and disable CFG at the first step. We found that guidance should be enabled during the early steps to support structural edits. Our method does not rely on blend words or any additional optimization techniques. All baseline methods were run with their default settings as provided by the authors or official implementations.

Figure 5: Examples of image editing results obtained using our method with guidance and other approaches. The proposed method provides fine-grained control over the trade-off between applying edits and preserving image content.

## 5.1 IMAGE INVERSION

We compare our method with iCD (Starodubcev et al., 2024), GNRi (Samuel et al., 2025), DDIM inversion (Song et al., 2022) and ReNoise SDXL-Turbo (Garibi et al., 2024) on the image reconstruction task.

**Qualitative evaluation** We find that our method performs significantly better on well-defined Pie-Bench prompts. A subset of our results, obtained by iCD backbone, is shown in Figure 4, where our approach outperforms other methods, including full-step DDIM, in terms of structural consistency and detail preservation (see Appendix A.3 for more examples with other backbones).

**Quantitative evaluation** We evaluate image reconstruction quality using mean-squared error (MSE), ImageReward and LPIPS. In Table 1 and in Figure 6, we demonstrate that our method outperforms all competing methods and even full-step DDIM inversion, demonstrating that the result is precisely accurate for an accelerated approach.

## 5.2 TEXT-GUIDED IMAGE EDITING

To validate our approach, we compare it with leading fast methods (iCD Starodubcev et al. (2024), TurboEdit Deutch et al. (2024), InfEdit (Xu et al., 2023b), ReNoise SDXL-Turbo Garibi et al. (2024)) and full-step diffusion-based methods (NTI Mokady et al. (2022), NPI (Miyake et al., 2024), Guide-and-Rescale (Titov et al., 2024)). Across all backbones, our method consistently outperforms accelerated methods in image fidelity, achieves performance on par with the strongest full-step baselines, and remains the fastest approach in our comparison.

**Qualitative evaluation** We present a subset of our results, obtained with iCD backbone, in Figure 5. As can be seen, our method enables precise edits while preserving the context and details of the original image. Despite the strong influence of the target prompt, ReNoise and TurboEdit exhibit a low level of content preservation, iCD outputs often contain artefacts and fail to maintain subject identity. InfEdit significantly reduces editability while strongly preserving the original image. Some images show no visible edits at all, while others exhibit incomplete or minimal changes (see Appendix A.4 for more examples with other backbones). For NPI and NTI, our approach provides more precise edits. Furthermore, it achieves results comparable to those of full-step diffusion-based models.

**Quantitative evaluation** We evaluate results using ImageReward (Xu et al., 2023a), DINOv2 (Oquab et al., 2024), LPIPS, and CLIP (Hessel et al., 2022). Most accelerated models tend

Table 1: Evaluation metrics for image reconstruction using our method and competing approaches on the MS-COCO validation set. Our method achieves higher reconstruction quality compared to accelerated baselines.

| Model | MSE ↓ | ImageReward ↑ | LPIPS ↓ |
|---|---|---|---|
| DDIM (SD) | 0.077 | 0.104 | 0.34 |
| iCD (SD) | 0.034 | -0.028 | 0.307 |
| Inverse-and-Edit (DreamShaper-LCM) | **0.017** | 0.304 | **0.225** |
| Inverse-and-Edit (iCD) | 0.021 | 0.344 | 0.228 |
| Inverse-and-Edit (SDXL-Turbo) | 0.049 | **0.515** | 0.34 |
| Renoise (SDXL-Turbo) | 0.1 | 0.367 | 0.402 |
| Inverse-and-Edit (Hyper SD) | 0.019 | 0.3 | 0.229 |

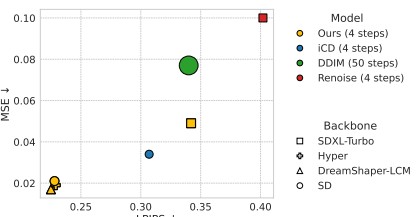

Figure 6: Quantitative evaluation on the MS-COCO validation set. Our method enables efficient and perceptually faithful image reconstruction, making it suitable for real-time applications.

Table 2: Metrics for edited images produced by our method and other baselines. Our approach significantly surpasses accelerated methods across all backbones in terms of image fidelity, while still supporting a high degree of editability. At the same time, method achieves results comparable to full-step baselines.

| Model | DINOv2↑ | LPIPS↓ | CLIP↑ | IR↑ | Sec. per edit |
|---|---|---|---|---|---|
| NPI (SD v1.5) | 0.632 | 0.302 | 0.302 | 0.22 | 9 |
| NTI (SD v1.5) | 0.795 | 0.25 | 0.294 | -0.03 | 115 |
| ReNoise (SD v1.5) | 0.504 | 0.446 | 0.315 | 0.36 | 9.19 |
| GaR (SD v1.4) | 0.721 | 0.277 | 0.307 | 0.25 | 19.73 |
| InfEdit (DreamShaper-LCM) | 0.781 | **0.236** | 0.298 | **0.16** | 3.48 |
| **Inverse-and-Edit (DreamShaper-LCM)** | **0.82** | **0.236** | **0.3** | **0.16** | 0.6 |
| TurboEdit (SDXL-Turbo) | 0.663 | 0.358 | 0.307 | **0.54** | 1.04 |
| GNRi (SDXL-Turbo) | **0.685** | 0.394 | 0.298 | 0.2 | 0.6 |
| ReNoise (SDXL-Turbo) | 0.561 | 0.426 | 0.307 | 0.37 | 3.09 |
| **Inverse-and-Edit (SDXL-Turbo)** | 0.682 | **0.35** | **0.309** | **0.54** | 0.6 |
| iCD (SD v1.5) | 0.701 | 0.323 | 0.302 | 0.1 | 1.05 |
| **Inverse-and-Edit (iCD)** | **0.719** | **0.312** | **0.304** | **0.31** | 0.57 |
| **Inverse-and-Edit (Hyper-SD)** | 0.68 | 0.34 | 0.302 | 0.12 | 0.59 |

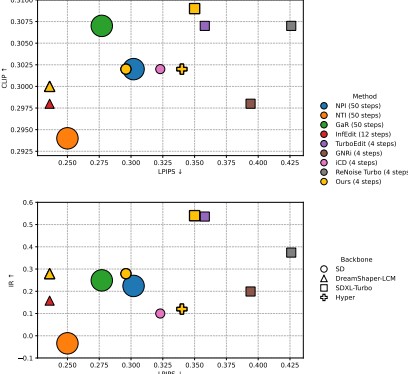

Figure 7: Quantitative evaluation on the Pie-Bench dataset. Our approach achieves similar results to full-step methods with less inference time.

to achieve stronger edit impact at the cost of content preservation. Elevated DINOv2 cosine distances and LPIPS metrics reflect the difficulty accelerated diffusion models encounter in preserving structural and visual detail. Our methods outperform all accelerated approaches across backbones in preserving image content and editing performance, achieving results comparable to full-step methods. In comparison with full-step methods, our method outperforms NPI and NTI, and achieves results comparable to Guide-And-Rescale. While the CLIP is lower, ImageReward is higher, showing that the edits are more aligned with human preferences despite being less favored by CLIP-based evaluation. Our LPIPS scores are higher, while DINOv2 cosine similarity is better. This suggests that our method preserves semantic and structural content more effectively, even though perceptual similarity appears lower. Overall results we present in Table 2 and in Figure 7.

Our approach demonstrates strong performance across both settings, enabling a smooth trade-off between fidelity and content preservation.

## 6 CONCLUSION

We propose a novel approach to training lightweight adapters that leverages the prior knowledge of arbitrary accelerated models for image-editing tasks. We show that simple end-to-end optimization over the entire image reconstruction process in accelerated diffusion models is effective, and we introduce a broad set of techniques for fast editing.

**Limitations** Our approach requires additional fine-tuning iterations and is data-dependent. Since the LPIPS backbone is trained to operate in pixel space, the proposed approach requires additional backpropagation through a VAE decoder, which increases the overall computational cost of optimization.

## 7 REPRODUCIBILITY

Details required to reproduce our method can be found in Method section 4, Experiments section 5 and in the Appendix A.2.

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

# A APPENDIX

## A.1 FINE-TUNE SETUP

We utilize LPIPS as the cycle-consistency loss since other latent-based variants (Huber, L2) do not perform well and often result in structural and visual mismatches. Our hyperparameters slightly differ across backbones: the learning rate is fixed to $1e - 6$ and the forward preservation coefficient to 1.5. We also match schedules to make $F_\psi$ symmetrical to $G_\theta$. For **DreamShaper-LCM** and **SDXL-Turbo** we choose the same target modules: to_q and to_v with different ranks: $r = 1$ for DreamShaper-LCM and $r = 32$ for SDXL-Turbo. The cycle-consistency coefficient is set to 0.5. Both models reach convergence after 3500 steps. For the **Hyper-SD** model we initialize $f_\psi$ and $g_\theta$ from an SDv1.5 LoRA-checkpoint with 4 steps and internal guidance, then fine-tune all inner LoRA-weights for 1000 steps with the cycle-consistency loss set to 0.05. For the **iCD** model we initialize $f_\psi$ and $g_\theta$ from publicly available checkpoints of forward and backward consistency models, which were distilled jointly. The cycle-consistency loss coefficient is set to 0.5.

Fine-tuning is performed using four H100 GPUs. Classifier-free guidance (CFG) is disabled during fine-tuning in order to preserve the model's ability to respond sensitively to editing operations.

## A.2 EDITING AND INVERSION SETUP

For inversion we disable CFG for all steps and use the source prompt for inversion and generation processes. We apply the procedure, which discussed in Section 4.3. Additionally, we employ a classifier-free guidance (CFG) (Ho & Salimans, 2022), which strengthens adherence to the prompt by::

$$\hat{\varepsilon}_\theta(z_t, t, y) = \varepsilon_\theta(z_t, t, \varnothing) + \omega \cdot (\varepsilon_\theta(z_t, t, y) - \varepsilon_\theta(z_t, t, \varnothing)) \tag{9}$$

For SDXL-Turbo $\omega$ is set to 4.0, for Hyper-SD $\omega$ is set to 2.5, for DreamShaper-LCM we set $\omega$ to 2.2 and disable it for the first step. For the iCD we use dynamic CFG schedule since it is guidance distilled model. Specifically, we start with zero CFG at the first step, increase it to 7 at the second step, to 11 at the third step, and to 19 at the final step. We found that guidance should be enabled during the early steps to support structural edits. However, a high level can result in supersaturated images.

## A.3 RECONSTRUCTION RESULTS

We present examples of image reconstruction by the proposed approach in Figure 8, which demonstrate that our backbones consistently produce accurate and visually faithful results. The method remains robust in preserving details and background structure across a variety of image types.

## A.4 EDITING RESULTS

We present results separately for SDXL-Turbo in Figure 9 based models and for other approaches in Figure 10. The edited images highlight the effectiveness and the scalability of the proposed framework for accurate and semantically consistent edits that respect the original structure and context. Although Hyper-SD does not outperform all baselines on quantitative metrics, it produces visually high-quality images in comparison to other methods.

## A.5 ABLATIONS

**Editing.** As shown in Figure 11, our method does not require an additional P2P framework compared to the iCD approach and achieves good results while better preserving the original content.

**Fine-tune.** As shown in Table 3, our method significantly improves reconstruction on the validation set while maintaining stronger image-generation metrics for the DreamShaper-LCM and Hyper-SD backbones.

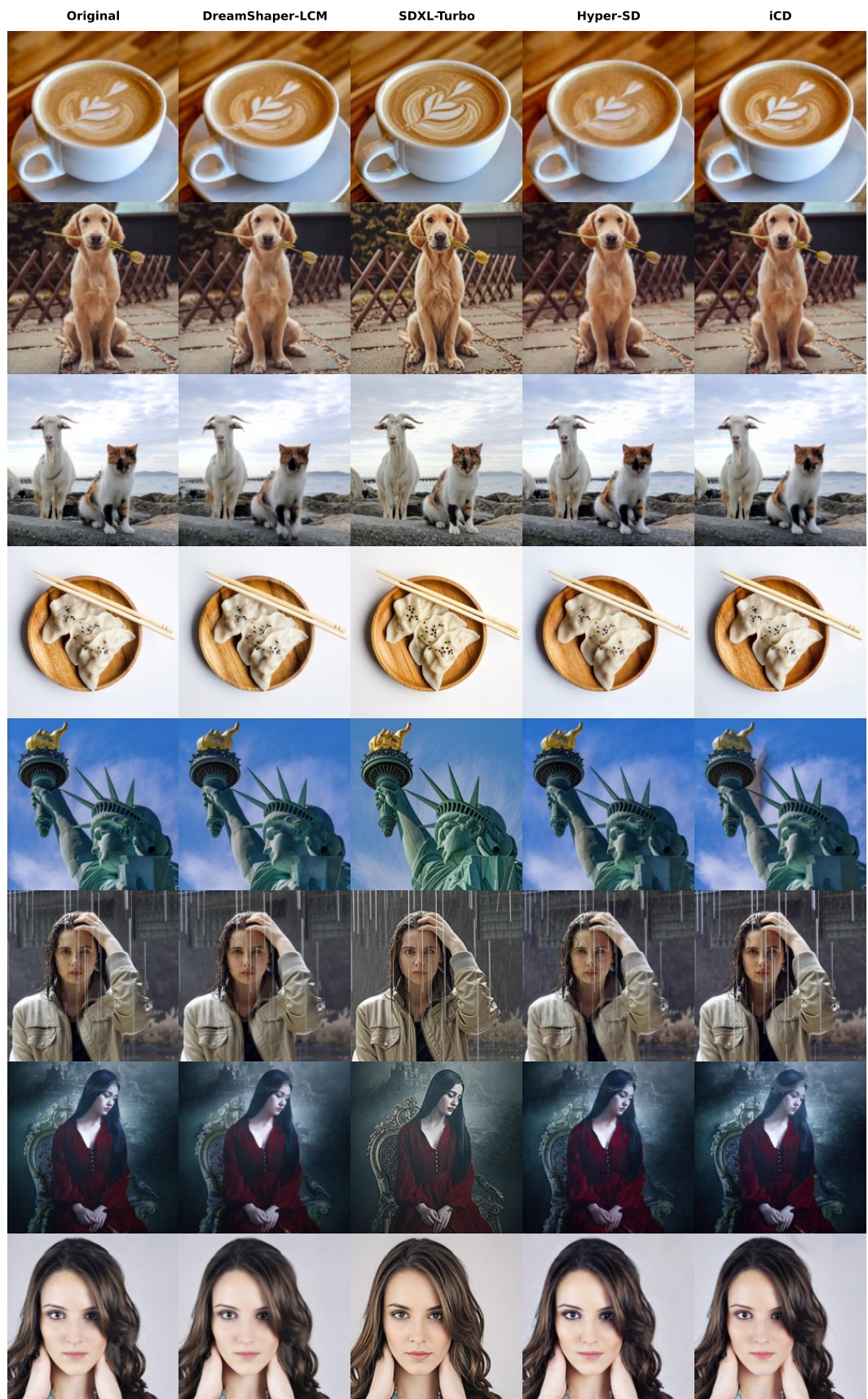

Figure 8: Examples of image reconstruction obtained using our method for different backbones.

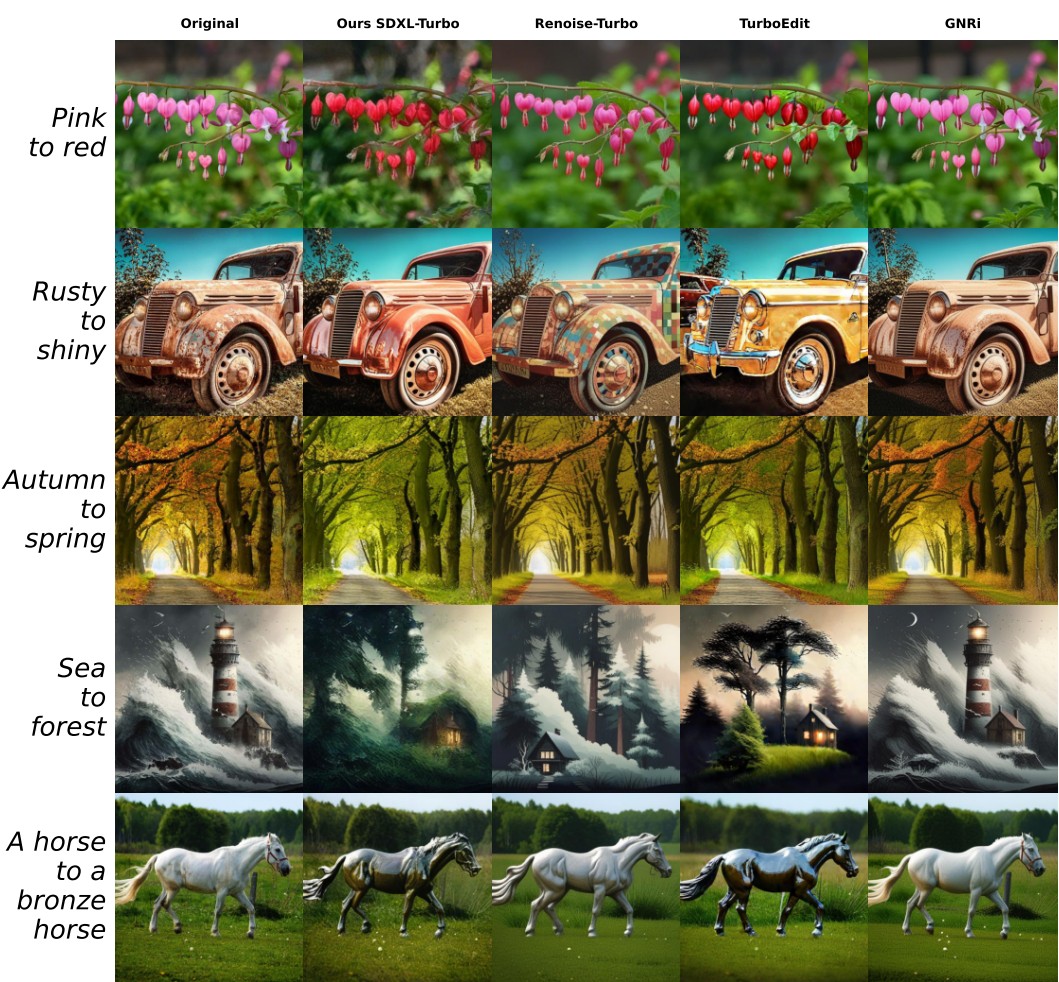

Figure 9: Examples of image editing results obtained using our method based on the SDXL-Turbo based backbone and other approaches.

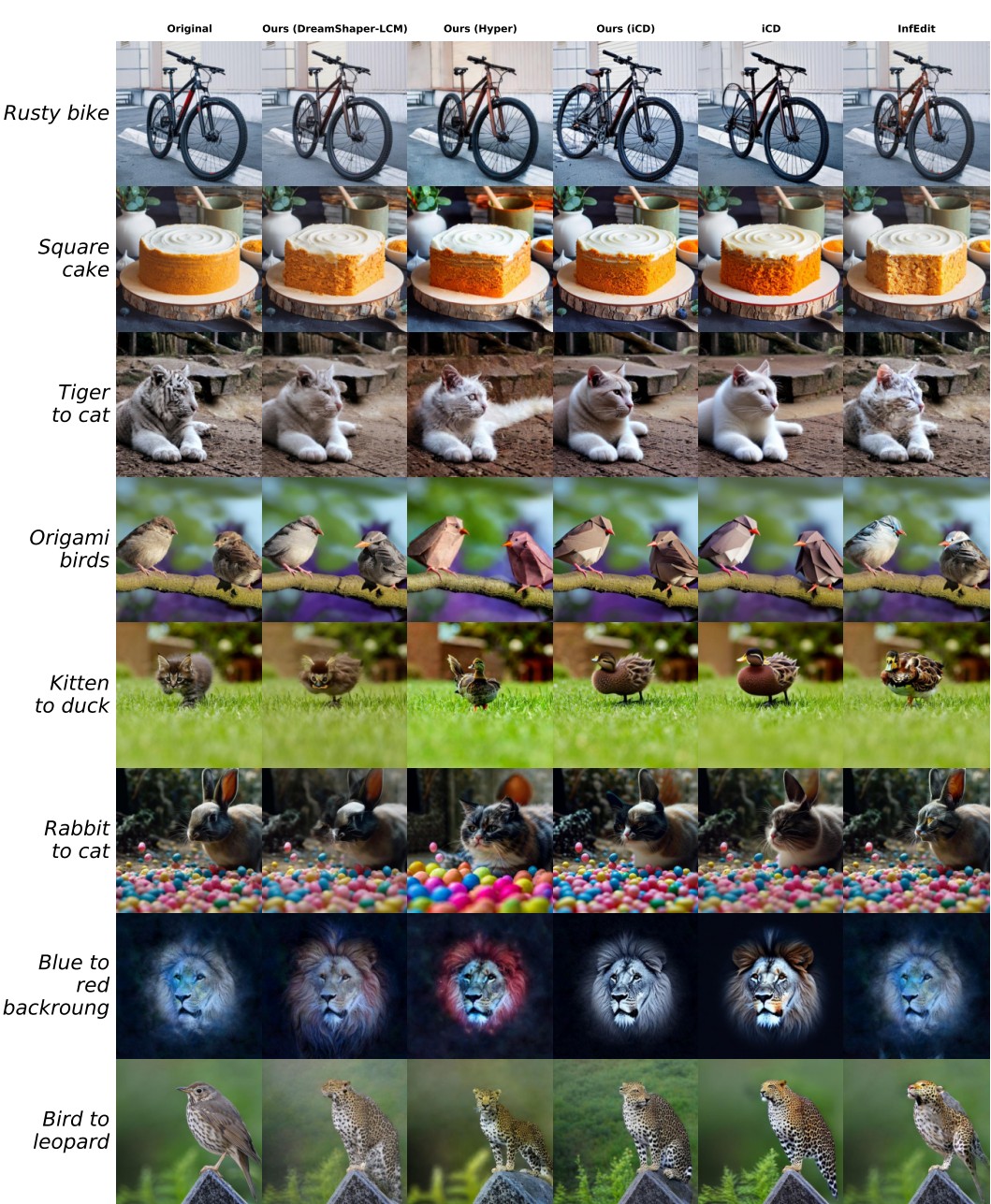

Figure 10: Examples of image editing results obtained using our method based on the SD based backbone and other approaches.

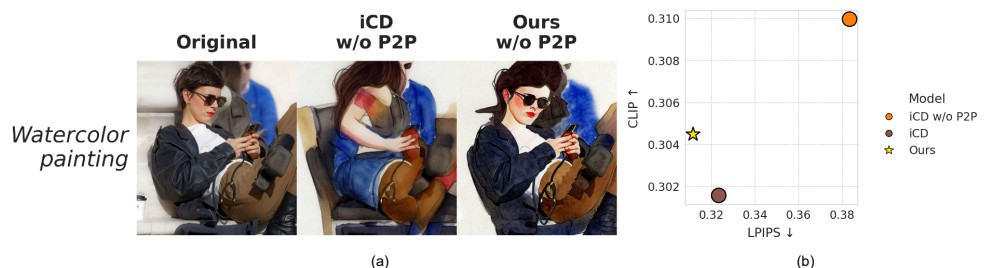

(a)                                                    (b)

Figure 11: (a): Visual comparison of results produced by our fine-tuned model based on iCD method and the baseline. While P2P is primarily required for iCD to preserve the input image, our method maintains high editing quality without relying on additional components. (b): Quantitative evaluation of the editing results. Our method demonstrates clear improvement with respect to image aesthetics and perceptual similarity.

| Method | CLIP ↑ | IR ↑ | MSE ↓ |
|---|---|---|---|
| *DreamShaper-LCM* | | | |
| iCD without cycle-consistency | 0.21 | -0.92 | 0.035 |
| Ours | 0.24 | -0.25 | 0.03 |
| *Hyper-SD* | | | |
| iCD without cycle-consistency | 0.21 | 0.26 | 0.09 |
| Ours | 0.3 | 0.03 | 0.008 |

Table 3: Ablation of suggested methods of fine-tune

