# OpenReview forum: "Inverse-and-Edit: Simple and Effective Framework for Fast Image Editing"
_ICLR.cc/2026/Conference — ICLR 2026 Conference Withdrawn Submission_

### Official Review · Reviewer_nb8L · 2025-10-17

**Soundness:** 3
**Presentation:** 3
**Contribution:** 2
**Rating:** 6
**Confidence:** 4

**Summary:**

The paper presents a method for image editing using accelerated diffusion models. The method builds on a previous work that trained a forward diffusion model with a consistency loss adapted for the forward process. In this paper, the authors add a cycle-consistency loss, improving reconstruction accuracy and quality. In this loss, gradients are propagated through all diffusion steps.

The authors demonstrate that the method works across a wide range of backbone models. It is compared against relevant baselines in terms of both reconstruction and editing. To ensure a fair comparison, each baseline is evaluated against the proposed method implemented with the same backbone.

**Strengths:**

- The paper shows plausible results. The inversion achieves accurate and high-quality reconstructions, the edits are successful and preserve the original image, and the method runs very efficiently. It reaches results comparable to those obtained with a full denoising process, but in significantly less time.
- The idea of adding a consistency loss is interesting and appears to contribute to the method’s success.
- Results are demonstrated across multiple backbones, each distilled into an accelerated model in a different way, showing the generalizability of the approach.

**Weaknesses:**

- The ablation studies are not comprehensive. For example, is the choice of LPIPS crucial for the success of the method? What happens if you use L2 instead, or in addition? In Section 4.2, the choice of which parameters to train in each configuration is not entirely clear and is not ablated. Also, it would be better to place the ablation studies in the main paper rather than the appendix.
- The presentation of iCD and the differences from it are not clear. It would be helpful to present it more fully in the preliminaries section and to discuss the differences between the methods more comprehensively.
- There are more advanced full-step editing methods, and it would be helpful to compare against them to better assess performance relative to full-step editing.
- The technical contribution over iCD is not very big, but I think it is enough.

**Questions:**

- Does the method work for flow models? For example, did you try it with Flux Schnell? If not, why doesn't it work?
- What are the memory requirements of the method? I assume it is pretty heavy as you need to propagate gradients through multiple application of the model.

---

### Official Review · Reviewer_KkLk · 2025-10-20

**Soundness:** 3
**Presentation:** 3
**Contribution:** 2
**Rating:** 4
**Confidence:** 4

**Summary:**

The paper introduces “Inverse-and-Edit,” a framework for fast, text-guided image editing on distilled diffusion backbones. The core idea is to train lightweight forward and backward consistency models (via LoRA adapters) end-to-end over the entire inversion and generation cycle, using a patch-wise LPIPS reconstruction loss combined with standard consistency-distillation and preservation terms, so that inversion becomes accurate enough to enable high-quality edits in a few steps. The method is designed to be backbone-agnostic and is instantiated on DreamShaper-LCM, SDXL-Turbo, Hyper-SD, and iCD. On PIE-Bench and MS-COCO, the approach reports strong reconstruction fidelity and competitive edit quality with sub-second latency per edit, often outperforming other accelerated editors while approaching full-step baselines.

**Strengths:**

* **Quality.** The framework is specified with clear objectives and segment-wise schedules; training regimes are adapted per backbone (e.g., joint vs. partial fine-tuning), and editing then reduces to a single noising→denoising pass without auxiliary modules (Prompt-to-Prompt/MasaCTRL).
* **Empirical results.** On MS-COCO inversion, the method improves MSE/LPIPS vs. iCD/ReNoise/DDIM; on PIE-Bench editing, it surpasses other accelerated methods and is comparable to strong full-step baselines, with reported per-edit latency around 0.6s on several backbones.
* **Breadth/compatibility.** Demonstrations span generator-based and consistency-based backbones (DreamShaper-LCM, SDXL-Turbo, Hyper-SD, iCD), supporting the claim of backbone-agnostic applicability.

**Weaknesses:**

* **Scope and external validity.** Most evaluation relies on PIE-Bench and automatic metrics; there is no user study, and the analysis of hard edit types (large geometry/viewpoint changes, compositional edits) is limited, making it hard to assess perceptual advantages beyond reported scores.
* **Training cost and practicality.** Despite fast inference, the approach needs extra fine-tuning (including backprop through the VAE due to pixel-space LPIPS), which introduces computational overhead and potential data dependence; practical guidance on compute budget and overfitting is sparse.
* **Backbone-specific knobs.** The framework still requires per-backbone choices (e.g., which modules/ranks to tune; deterministic adaptation for DreamShaper-LCM; CFG schedules), which may hinder true plug-and-play deployment without careful re-tuning.

**Questions:**

* Can the authors provide wall-clock fine-tuning cost (GPU hours, dataset size, epochs) per backbone, and guidance for practitioners seeking a “small-compute” setting that still achieves most of the gains?
* How robust is the method to CFG schedules and step counts across backbones? Please include quality-vs-steps and quality-vs-CFG results, ideally on PIE-Bench subsets with structural vs. stylistic edits.
* For SDXL-Turbo/Hyper-SD, which modules/ranks are most critical to tune, and can a single set of LoRA hyperparameters generalize across datasets (e.g., COCO→PIE-Bench) without re-tuning? A small cross-dataset generalization table would be convincing.

---

### Official Review · Reviewer_bjKj · 2025-10-29

**Soundness:** 2
**Presentation:** 1
**Contribution:** 1
**Rating:** 2
**Confidence:** 2

**Summary:**

This paper addresses image editing through diffusion inversion. The core contribution is a modified training procedure for iCD (Invertible Consistency Distillation) that involves backpropagating the LPIPS loss through the entire generation process, rather than just a single step.

**Strengths:**

* The method demonstrates improved latency and better performance on downstream metrics compared to the reported baselines.
* The core idea of backpropagating the loss through the entire generation process is a logical and intuitive approach to improving inversion quality and subsequent editing performance.

**Weaknesses:**

* The primary contribution appears to be an improvement to the iCD training regime. However, the novelty of this modification is not clearly established.
* The qualitative results presented in the figures often appear visually similar to those produced by the iCD baseline, making the practical benefits of the proposed method less apparent.
* The paper's clarity could be significantly improved. I found it difficult to fully grasp the precise technical differences between the proposed method and the original iCD. Section 4.1, in particular, should be revised to explicitly and clearly delineate these differences.

**Questions:**

1.  **L212 vs. Sec 4.2:** L212 states that iCR requires retraining bidirectional models, unlike the proposed method. However, Section 4.2 seems to indicate that a separate model is trained for each diffusion model. Could the authors please clarify this apparent contradiction?
2.  **Table 2 (Quantitative Results):** Table 2 compares the proposed method (trained on the COCO train set) against other baselines using the COCO validation set. For a fair comparison, were the baselines (such as iCD) also trained on the COCO train set?
3.  **Figure 1:** Figure 1 provides inversion latencies, showing the proposed method is faster. Are these latencies image-dependent, or are they consistent across different inputs (e.g., dependent only on image resolution)?
4.  **L223 & L272:** L223 claims the approach avoids "stochastic samplers, internal solver access, or joint distillation." What are the practical implications of this? Furthermore, L272 mentions the method *does* "jointly optimize both models." This seems to conflict with the claim of avoiding "joint distillation." Please clarify this.
5.  **L309:** L309 states that no post-training editing framework (like Prompt-to-Prompt) is required. Does this also apply to the baselines, such as iCD? If not, what specific editing framework was used to generate the baseline results in the experiments?
6.  **Training Cost:** Given that the method requires backpropagation through the entire generation process, what are the implications for training time and VRAM/memory usage compared to methods that do not?

---

### Official Review · Reviewer_CMGs · 2025-11-01

**Soundness:** 3
**Presentation:** 3
**Contribution:** 2
**Rating:** 4
**Confidence:** 4

**Summary:**

This paper proposes Inverse-and-Edit, a unified framework for fast and high-quality text-guided image editing across various accelerated diffusion backbones. The key idea is to train forward and backward consistency models via a cycle-consistency loss to improve inversion fidelity, allowing the system to perform precise edits in only a few steps. The method shows strong reconstruction quality and state-of-the-art performance in fast editing scenarios.

**Strengths:**

- The method consistently outperforms other accelerated editing baselines and achieves performance comparable to full-step diffusion editing methods, both quantitatively and qualitatively.
- The approach is compatible with multiple distilled diffusion backbones without requiring architectural modifications, which enhances its practicality and general applicability.
- The method supports fast editing at inference time, enabling efficient real-time or interactive usage.

**Weaknesses:**

- At a high level, the approach resembles iCD, with the primary difference being the fine-tuning strategy and inclusion of a cycle-consistency reconstruction loss. Conceptually, the contribution may be seen as an incremental extension rather than a fundamentally new inversion or editing formulation.
- Unlike many recent training-free editing pipelines, this framework requires additional fine-tuning, which increases computational cost and limits accessibility.
- While the method achieves improvements in reconstruction and editing quality, the paper does not deeply analyze failure modes or conditions under which the method may underperform.

**Questions:**

- Recent open-source editing models such as Flux Kontext or Qwen-Image-Edit would serve as more relevant baselines. Even though their architectures differ, comparing editing quality against these models would better contextualize the practical impact of the proposed approach.
- A discussion of situations where the model fails or performs poorly would strengthen the paper.

---

### Note · Authors · 2025-11-14

I have read and agree with the venue's withdrawal policy on behalf of myself and my co-authors.